# A Case of Penile Metastasis from Prostate Cancer, Identified by ^68^Ga-PSMA PET/CT, Mimicking Peyronie’s Disease: A Diagnostic Challenge

**DOI:** 10.3390/diagnostics13152509

**Published:** 2023-07-27

**Authors:** Farid Gossili, Niels Christian Langkilde, Helle D. Zacho

**Affiliations:** 1Department of Nuclear Medicine, Clinical Cancer Research Center, Aalborg University Hospital, 9000 Aalborg, Denmark; h.zacho@rn.dk; 2Department of Clinical Medicine, Aalborg University, 9000 Aalborg, Denmark; 3Department of Urology, Aalborg University Hospital, 9000 Aalborg, Denmark; ncl@rn.dk

**Keywords:** ^68^Ga-PSMA PET/CT, prostate cancer, penile metastases, Peyronie’s disease

## Abstract

A 70-year-old man with high-risk prostate cancer (PCa) received radiation therapy and androgen deprivation therapy (ADT). The patient developed penile tenderness, compatible with Peyronie’s disease upon physical examination. An ultrasound revealed a matching hypoechoic plaque and a thrombus in the vena dorsalis profunda, which were treated with anticoagulants. A follow-up ultrasound showed no abnormalities. Despite the use of analgesics, the patient suffered from persistent pain, later accompanied by an increasing PSA level of up to 7.5 ng/mL, despite ADT. ^68^Ga-PSMA PET/CT showed a PSMA uptake consistent with PCa penile metastasis. Due to severe pain and the presence of metastatic PCa, the patient was referred for penectomy. Histopathological analysis confirmed metastases originating from the PCa. This case underscores the importance of ^68^Ga-PSMA PET/CT in diagnosing PCa metastases and vigilance towards urogenital symptoms as potential indicators of metastases, despite the rarity of penile metastases.

**Figure 1 diagnostics-13-02509-f001:**
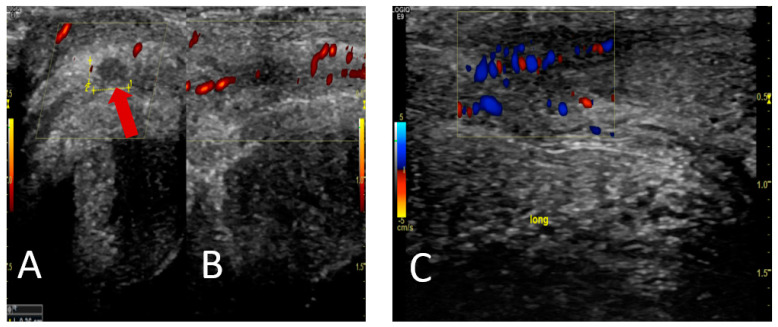
A 70-year-old male patient was diagnosed with high-risk prostate cancer (Gleason score was 9 (4 + 5), PSA 42 ng/mL, and cT3). There were no metastases according to the primary CT and bone scintigraphy. The patient underwent curatively intended radiotherapy combined with androgen deprivation therapy (ADT) for three years. Subsequently, the patient presented with intermittent pain and tenderness in the penis, which sometimes appeared spontaneously or was completely absent. The physical examination yielded negative findings for phimosis, while the palpation of both corpus cavernosa revealed the presence of two plaques, raising suspicion of Peyronie’s disease. Peyronie’s disease is a benign chronic lesion that exhibits an age-dependent incidence pattern, with the highest occurrence observed during the fifth decade of life; it results from fibrosis of the connective tissue of the penis, leading to fibrous scars, plaques, or calcifications and loss of elasticity, with clinical signs including penile pain, penile sclerosis, painful erections, erectile dysfunction, curvature, and penile deformity [1,2]. Among the various complaints associated with penile discomfort, penile pain stands out as a prevalent issue, affecting a significant proportion of patients during the initial stages and usually resolves within 12 months [2]. The patient was advised to take painkillers and wait for the tenderness to disappear within six months to a year. A penile ultrasound scan showed a hypoechoic plaque proximal to the glans penis, corresponding to palpable filling on the right side dorsally, as indicated by the red arrow (**A**), and partially recanalized thrombosis of the vena dorsalis profunda with adjacent small oedema (**B**). The patient was prescribed anticoagulant medication, and a thrombosis check was conducted after one month. The follow-up ultrasound showed a decrease in oedema/reaction and thrombus, indicating incipient flow through the area, and it did not show any hypoechoic zones (**C**).

**Figure 2 diagnostics-13-02509-f002:**
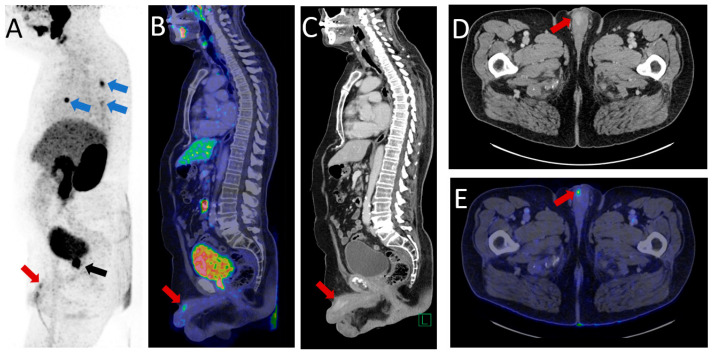
The patient had persistent pain in the penis despite the use of analgesics, particularly at the tip of the penis. A prostate-specific antigen (PSA) test was performed, revealing an initial level of 2.8 ng/mL (nadir 0.8 ng/mL), followed by a rise in PSA levels to 7.5 ng/mL within three months, despite the patient undergoing reinitiation of ADT. Following a multidisciplinary team assessment, it was determined that the patient had castration-resistant prostate cancer, and the patient was referred for prostate-specific membrane antigen (PSMA) PET/CT. The ^68^Ga-PSMA PET/CT demonstrated intense PSMA uptake in the prostate, indicating local recurrence, as shown by the black arrow on the maximum intense projection (MIP) (**A**). Furthermore, a distinct area of intense PSMA uptake, correlating with contrast enhancement on the CT scan, was observed on the right side of the penis, indicated by the red arrows on the MIP (**A**), fused sagittal PET/CT (**B**), sagittal contrast-enhanced CT (**C**), contrast-enhanced axial CT (**D**), and fused axial PET/CT (**E**), suggesting a metastasis from the prostate cancer. Moreover, no metastases were seen in the lymph nodes or bones; however, high PSMA uptake was present in the nodules in both lungs (blue arrows on the MIP (**A**)). Given the increasingly severe pain and the high probability of metastatic prostate cancer, the patient was referred for penectomy for pain relief. Although there was a theoretical risk that the patient was suffering from Peyronie’s disease rather than cancer, he proceeded to undergo surgery, and subsequent histopathology revealed an adenocarcinoma originating from the prostate. It should be noted that metastases from prostate cancer typically occur in the lymph nodes and bones, with lung and lever metastases being less common [3] and penile metastases being exceedingly rare (0.1–0.5%) [4,5]. Studies have demonstrated that penile metastases are a rare and typically late manifestation of metastatic prostate cancer, with only a very small percentage of cases presenting with the penis as the initial site of metastasis [6,7]. Clinical presentation in this patient group frequently includes painless nodules in the corpus cavernosum or glans, penile skin changes, abnormal penile erection and priapism, urinary retention, pelvic, perineal, and penile pain, dyspareunia, and haematuria [7]. Therefore, penile metastases must be carefully considered as an important differential diagnosis of Peyronie’s disease; however, misdiagnosis or misinterpretation of penile metastases as Peyronie’s disease has occurred [8,9,10,11,12]. Most reported cases involve metastases from cancers other than prostate cancer [9,10,11,12]. An instance of reported penile metastases originating from prostate cancer, initially misdiagnosed as Peyronie’s disease, was observed during the primary staging of the disease and distinguished by a significant dissemination of malignancy [8]. In the current clinical scenario, the utilization of penile ultrasound failed to identify any metastatic lesions, leading to an erroneous diagnosis of cavernous sclerosis and supporting the clinical suspicion of Peyronie’s disease. Ultrasound imaging does not provide substantial benefits in accurately characterizing penile tumours, as confirmed by the literature [6]. Conversely, PSMA PET/CT imaging has emerged as a noninvasive modality for the detection of penile metastases [13]. This case substantiates the significance of employing PSMA PET/CT imaging as a modality in the diagnostic evaluation of prostate cancer metastasis, alongside the necessity to exercise diligent vigilance towards urogenital symptoms exhibited by prostate cancer patients. Even though penile metastases are infrequent, these symptoms may serve as pivotal indicators of metastatic dissemination, distinct from alternative pathological conditions.

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
