# Peer review of "A Case of Penile Metastasis from Prostate Cancer, Identified by 68Ga-PSMA PET/CT, Mimicking Peyronie’s Disease: A Diagnostic Challenge"

_diagnostics, 2023, doi:10.3390/diagnostics13152509_

Round 1

Reviewer 1 Report

The authors provided a full case of penile metastasis from PCa.

The style and the scientifical sound are good.

However, I might think that non-urologist/andrologist may find some difficulties in following the case and I suggest to better describe the differential diagnosis with Peyronie's disease by better describing the clinical appearance as in PMID: 36426559.

Reviewer 2 Report

This manuscript reports a case of penile metastasis from prostate cancer identified by PET/CT, which was compatible with Peyronie’s disease. 

Previous studies have reported cases with penile metastasis from prostate, some have also discussed the differentiation betweenpenile metastasis and Peyronie's disease. But some studies discussing the penile metastasis are not cited and discuss in this manuscript. This study does not provide a comprehensive comparison with previous research to emphasize its distinctiveness and significance.

None.

Reviewer 3 Report

A well-written short text on a relatively rare location of prostate cancer metastases. I believe that this manuscript should be accepted for publication in the journal Diagnostics as an Interesting Images article type.

Round 2

Reviewer 2 Report

I believe the revision has been properly revised and meets the publication standard.